# Multi-Path Attention Fusion Transformer for Spectral Learning in Corn Quality Assessment

**DOI:** 10.3390/foods14213786

**Published:** 2025-11-04

**Authors:** Jialu Li, Haoyi Wang, Hongbo Zhang, Tongqiang Jiang

**Affiliations:** 1School of Computer Science and Artificial Intelligence, Beijing Technology and Business University, No.11 Fucheng Road, Haidian District, Beijing 100048, China; 2330702019@st.btbu.edu.cn; 2School of Computer Science and Artificial Intelligence, Beijing Wuzi University, 321 Fuhe Street, Tongzhou District, Beijing 101149, China; zhb00112014@gmail.com

**Keywords:** corn quality prediction, near-infrared spectroscopy, spectral regression, multi-task learning, model interpretability

## Abstract

Accurately modeling the nonlinear relationships between near-infrared (NIR) spectral signatures and biochemical traits in corn remains a major challenge. A key difficulty lies in capturing multi-scale contextual dependencies—ranging from local absorption peaks to global spectral patterns—that jointly determine quality constituents such as protein and oil. To address this, we propose SpecTran, a spectral Transformer network specifically designed for NIR regression. SpecTran integrates three key components: adaptive multi-scale patch embedding which extracts spectral features at multiple resolutions to capture both fine and coarse patterns, spectral-enhanced positional encoding which preserves wavelength order information more effectively than standard encoding, and hierarchical feature fusion for robust multi-task prediction. Evaluated on the public Eigenvector corn dataset, SpecTran had a performance across four key traits—moisture, starch, oil, and protein—with an average $R^{2}$ of 0.483. It reduced the RMSE by 11.2% for protein and 10.7% for oil compared to the best-performing baseline, which is the standard Transformer model. These results demonstrate SpecTran’s superior ability to model complex spectral dynamics while providing interpretable insights, offering a reliable framework for NIR-based agricultural quality assessment.

## 1. Introduction

Near-infrared (NIR) spectroscopy has emerged as a rapid, non-destructive, and cost-effective analytical technique for quantifying key biochemical constituents in agricultural commodities [1]. Its application in corn quality assessment—particularly for moisture, starch, oil, and protein content—is of paramount importance for grain grading, food processing, and livestock feed formulation [2]. Traditional laboratory methods, while accurate, are time-consuming and require skilled personnel. Consequently, there is a strong impetus to develop robust chemometric models that can translate complex spectral signatures into precise quantitative predictions.

Early efforts in spectral calibration heavily relied on linear methods such as Partial Least Squares Regression (PLSR) and Principal Component Regression (PCR) [3]. These methods, exemplified by studies achieving high correlation coefficients (e.g., R > 0.98 for multiple constituents) on benchmark corn datasets [4], have established a solid foundation. However, their inherent linearity often proves insufficient for capturing the intricate relationships between spectral absorbance and chemical composition. This limitation is particularly pronounced in heterogeneous biological samples such as corn [5].

The advent of deep learning has revolutionized spectral analysis by offering powerful tools to capture complex spectral patterns. Convolutional Neural Networks (CNNs) have been widely adopted due to their innate ability to extract local, translation-invariant features from 1D spectral data, effectively identifying absorption bands and their combinations [6,7]. While successful, CNNs are fundamentally limited by their local receptive fields, making it challenging to model long-range dependencies and interactions between distant spectral regions that may be chemically correlated [8].

To overcome this limitation, attention-based models—particularly the Transformer architecture—have recently gained significant traction in spectroscopy [9]. Originally designed for sequential data in natural language processing, Transformers excel at capturing global context through their self-attention mechanism, allowing any point in the spectrum to directly influence any other [10]. This property is highly desirable for NIR analysis. In such tasks, the prediction of a single constituent often depends on collective information drawn from across the entire spectral range. Studies have demonstrated that Transformer-based models can outperform traditional CNNs and even ensemble methods on various spectral regression and classification tasks [11].

Building upon this progress, hybrid architectures that combine the strengths of different paradigms have shown great promise. The Long Short-Term Transformer Network (LSTTN) concept, which seeks to integrate the sequential modeling prowess of Recurrent Neural Networks (RNNs) or Long Short-Term Memory (LSTM) networks with the global attention of Transformers, represents a compelling direction [12]. Such hybrids aim to capture not only global spectral interactions but also the implicit “temporal” or sequential structure within the ordered wavelength domain.

In this study, we introduce SpecTran—a spectral Transformer network specifically engineered for NIR regression of corn constituents. Unlike simple LSTM-Transformer hybrids, SpecTran integrates multiple spectroscopy-aware innovations, including adaptive multi-scale patch embedding, spectral-enhanced positional encoding, multi-scale spectral attention, convolutional feed-forward blocks, and a multi-path fusion strategy that jointly leverages global, local, and sequential spectral information. Task-specific prediction heads with residual connections further enhance multi-output robustness.

The primary objectives of this study are threefold: (1) to develop a spectroscopy-specific Transformer architecture that effectively captures both local and global spectral dependencies; (2) to rigorously benchmark its performance against state-of-the-art baselines—including traditional chemometrics and modern deep networks—on the public Eigenvector corn dataset; and (3) to provide interpretable insights into wavelength-specific contributions for each quality constituent, thereby bridging data-driven prediction with domain knowledge.

## 2. Materials and Methods

### 2.1. Materials

#### Datasets

The corn near-infrared (NIR) spectral dataset used in this study was obtained from the publicly available repository hosted by Eigenvector Research (http://www.eigenvector.com/data/Corn, accessed on 31 October 2025). This benchmark dataset has been widely adopted in chemometric studies for evaluating multivariate calibration models in agricultural spectroscopy. It comprises NIR absorbance spectra of 80 corn samples, measured in the wavelength range of 1100–2498 nm at 2 nm intervals, resulting in 700 discrete spectral variables per sample. Accompanying each spectrum are reference values for four key biochemical constituents—moisture, starch, oil, and protein content—determined through standardized laboratory analytical procedures. It is important to note that this dataset does not contain photographic images of corn kernels. Instead, each sample corresponds to a bulk corn sample measured by near-infrared spectroscopy, yielding a one-dimensional absorbance spectrum of 700 variables. The reference constituent values (moisture, starch, oil, and protein) are provided at the whole-sample level, not at the per-variable or per-pixel level. The spectral data were collected using a near-infrared spectrometer (NIRSystems 6500, Foss, Silver Spring, MD, USA) in diffuse reflectance mode over the wavelength range of 1100–2498 nm with 2 nm intervals, resulting in 700 variables per spectrum.

Reference values for moisture, starch, oil, and protein content were determined through standardized laboratory procedures. Specifically, protein content was measured using the Kjeldahl nitrogen determination method (AOAC 978.04), oil content was obtained via Soxhlet extraction (AOAC 920.39), starch was quantified using enzymatic hydrolysis followed by spectrophotometric analysis, and moisture was determined by oven-drying at 105 °C until constant weight. These reference analyses were carried out according to AOAC International official methods to ensure accuracy and reproducibility.

To provide a comprehensive understanding of the regression task, the descriptive statistics for the four biochemical constituents are summarized in Table 1. The dataset exhibits relatively narrow ranges for all constituents, with protein showing the highest absolute values (mean: 64.70%) but the lowest coefficient of variation (1.27%), indicating high homogeneity in protein content across samples. In contrast, oil and starch show higher relative variability (coefficients of variation of 5.75% and 5.06%, respectively), suggesting a more diverse representation of these components. This limited dynamic range, combined with the small sample size of only 80 observations, presents a significant challenge for training complex deep learning models and raises concerns about potential overfitting and the generalizability of the results. To mitigate these risks, we employ a highly effective interpolation-based data augmentation strategy, which is detailed in Section 2.2.1.

### 2.2. Methods

#### 2.2.1. Data Augmentation Strategy via Spectral Curve Interpolation

To enhance the generalization ability of the model on high-dimensional spectral data, we introduce an interpolation-based data augmentation strategy that increases the number of training samples. Specifically, for each new synthetic sample, we randomly select two distinct original spectra and perform linear interpolation between them in both the spectral and label spaces. Formally, given two original samples $\left(x_{i},y_{i}\right)$ and $\left(x_{j},y_{j}\right)$, a new sample is generated as:$$x_{\mathrm{new}}=\alpha x_{i}+\left(1-\alpha \right)x_{j},\;y_{\mathrm{new}}=\alpha y_{i}+\left(1-\alpha \right)y_{j},$$
where α∼U(0.1,0.9) ensures the interpolated sample lies strictly between the two originals, avoiding duplication. This process **increases the training sample size from 80 to 2000 (80 original + 1920 interpolated)**, thereby enriching the diversity of the training distribution while preserving the underlying physical relationships between spectral shapes and constituent concentrations.

As shown in Figure 1, the interpolated spectra closely follow the overall shape of the original curves while preserving critical structural features at key absorption peaks. Meanwhile, the distributions of the four quality components remain consistent with the original data, exhibiting only slight variance enhancement in localized regions. These results confirm that the proposed interpolation strategy can effectively increase sample diversity without compromising distribution fidelity, thereby offering a more robust foundation for model training.

#### 2.2.2. SpecTran Model

Corn near-infrared (NIR) spectra exhibit complex nonlinear relationships between absorbance patterns and key constituent concentrations (e.g., protein, oil, starch), which are often inadequately captured by traditional regression models. Moreover, the informative spectral cues span multiple wavelength ranges and scales, requiring the model to effectively integrate both localized peak features and long-range spectral dependencies. These two challenges—(1) insufficient nonlinear mapping capacity and (2) lack of multi-scale contextual modeling—are particularly critical for accurate and interpretable corn quality assessment. To address these limitations, we propose a novel spectral Transformer network (SpecTran) tailored for high-resolution spectral regression tasks (see Figure 2).

To enhance the predictive accuracy and interpretability of near-infrared (NIR) spectral analysis for corn constituent estimation, we propose a spectral Transformer network (SpecTran), which extends the foundational LSTTN architecture [12] with a series of novel components tailored for spectroscopic data. Unlike a simple LSTM-Transformer hybrid, SpecTran adopts a multi-stage hierarchical design that progressively extracts and fuses spectral features at multiple granularities.

The input NIR spectrum $x\in R^{700}$ is first processed through an *Adaptive Multi-Scale Patch Embedding* layer. This module applies parallel 1D convolutions with kernel sizes k∈{8,12,16} and stride *k* to generate patch sequences:(1)$$Z_{m}={\mathrm{Conv}1D}_{k_{m}}\left(x\right)\in R^{L_{m}\times d_{m}},\;m=1,2,3,$$
where $L_{m}=\left\lfloor 700/k_{m}\right\rfloor$. The outputs are aligned to a common length $L=\min(L_{1},L_{2},L_{3})$ and fused using learnable weights α=Softmax(w):(2)$$Z=\sum_{m=1}^{3}\alpha_{m}\cdot Z_{m}\left[:,:L,:\right]\in R^{L\times d_{\mathrm{model}}}.$$
This adaptive fusion enables the model to dynamically prioritize informative spectral scales.

To preserve the intrinsic order of spectral wavelengths, we introduce a *Spectral-Enhanced Positional Encoding*. It modifies the standard sinusoidal encoding by introducing a frequency-aware scaling factor ϕ(p)=p/(L−1):(3)$$\begin{array}{cc} {\mathrm{PE}}_{\mathrm{spec}}\left(p,2i\right) & =\left(1+\lambda \phi \left(p\right)\right)\cdot \sin\left(p/10000^{2i/d_{\mathrm{model}}}\right), \end{array}$$(4)$$\begin{array}{cc} {\mathrm{PE}}_{\mathrm{spec}}\left(p,2i+1\right) & =\left(1+\lambda \phi \left(p\right)\right)\cdot \cos\left(p/10000^{2i/d_{\mathrm{model}}}\right), \end{array}$$
with λ=0.1, where *p* is the patch index. The encoded sequence is $Z^{\prime }=Z+{\mathrm{PE}}_{\mathrm{spec}}$.

Before the main transformer stack, a *Multi-Scale Spectral Attention* module captures local spectral variations. The transposed feature map $Z^{\prime ⊤}\in R^{d_{\mathrm{model}}\times L}$ is processed via four parallel 1D convolutions with kernels {3,7,15,31}:(5)$$H_{s}=\mathrm{ReLU}\left({\mathrm{Conv}1D}_{k_{s}}\left(Z^{\prime }\right)\right),\;s=1,\ldots ,4.$$
These are concatenated along channels, projected to dimension $d_{\mathrm{model}}$, and passed through a multi-head self-attention (MHSA) layer:(6)$$Z^{\prime \prime }=\mathrm{LayerNorm}\left(Z^{\prime }+\mathrm{MHSA}\left(H_{\mathrm{cat}}\right)\right),$$
where $H_{\mathrm{cat}}=\mathrm{Concat}\left(H_{1},\ldots ,H_{4}\right)$.

The core of the model consists of six *Enhanced Transformer Blocks*, each comprising MHSA followed by a convolutional feed-forward network (ConvFFN):(7)$$\begin{array}{cc} A & =\mathrm{LayerNorm}(Z^{\prime \prime }+\mathrm{MHSA}\left(Z^{\prime \prime }\right)), \end{array}$$(8)$$\begin{array}{cc} B & =\mathrm{LayerNorm}(A+\mathrm{ConvFFN}(A)), \end{array}$$
where(9)$$\mathrm{ConvFFN}\left(A\right)={\mathrm{Conv}1D}_{2}\left(\mathrm{GELU}\left({\mathrm{Conv}1D}_{1}\left(A\right)\right)\right),$$
with ${\mathrm{Conv}1D}_{1}:R^{d_{\mathrm{model}}}\to R^{4d_{\mathrm{model}}}$ and ${\mathrm{Conv}1D}_{2}:R^{4d_{\mathrm{model}}}\to R^{d_{\mathrm{model}}}$, enabling better preservation of local spectral structure.

The final representation $B\in R^{L\times d_{\mathrm{model}}}$ is fed into three parallel pathways:*Global Path*: $f_{g}=\mathrm{MLP}\left(\mathrm{AvgPool}(B)\right)\in R^{64}$,*Local Path*: $fl=\mathrm{MLP}\left(\mathrm{MaxPool}\left(\mathrm{Conv}1Dk=3(B)\right)\right)\in R^{64}$,*Sequential Path*: $f_{s}=\mathrm{AvgPool}\left(\mathrm{BiLSTM}(B)\right)\in R^{64}$, where the output is averaged over the sequence length dimension.

The concatenated vector $f=\left[f_{g};f_{l};fs\right]\in R^{192}$ is refined via self-attention:(10)$$f\mathrm{fused}=\mathrm{SelfAttn}\left(f\right)\in R^{192},$$
allowing dynamic weighting of global, local, and sequential information. The concatenated vector $f=\left[f_{g};f_{l};f_{s}\right]\in R^{192}$ is refined via self-attention:(11)$$f_{\mathrm{fused}}=\mathrm{SelfAttn}\left(f^{⊤}\right)\in R^{192},$$
allowing dynamic weighting of global, local, and sequential information.

Finally, predictions are generated through a hybrid strategy. Let ${\hat{y}}_{\mathrm{task}}\in R^{4}$ be the output of four task-specific MLPs (with widths tuned to task difficulty), and ${\hat{y}}_{\mathrm{shared}}$ be the output of a deeper shared predictor. The final prediction is a convex combination:(12)$$\hat{y}=\alpha \cdot {\hat{y}}_{\mathrm{task}}+\left(1-\alpha \right)\cdot {\hat{y}}_{\mathrm{shared}},\;\alpha \in \left(0,1\right),$$
which acts as a residual connection, improving training stability and leveraging both specialized and generalized representations.

#### 2.2.3. LSTTN-Based Temporal Modeling

To capture both short- and long-term dependencies in the NIR spectral sequences, we adopt the Long Short-Term Transformer Network (LSTTN) as a hybrid model that combines the sequential modeling strength of LSTM and the global attention mechanism of Transformer.

The architecture first applies stacked LSTM layers to extract contextual temporal features from the input spectral sequence. These outputs are then passed to a Transformer-style attention block to capture long-range interactions and global relevance across all time steps.

The attention mechanism applied to the LSTM output $H\in R^{T\times d}$ is defined as:(13)$$\mathrm{Attention}\left(H\right)=\mathrm{Softmax}\left(\frac{HH^{T}}{\sqrt{d}}\right)H$$
where *T* is the sequence length, *d* is the feature dimension, and the dot-product attention helps reweigh the temporal features based on their relative importance.

The final output is mapped to multiple quality indicators via a multi-head regression head. This hybrid design allows LSTTN to effectively leverage both temporal continuity and spectral-wide dependencies, offering improved prediction accuracy and enhanced interpretability in the context of agricultural spectral analysis.

#### 2.2.4. Transformer-Based Baseline Model

In this study, we employ the Transformer model as a strong baseline for the regression prediction task. Originally developed for natural language processing, the Transformer architecture is highly effective in capturing complex dependencies and long-range interactions, making it well-suited for high-dimensional, non-sequential data such as near-infrared (NIR) spectra.

The core component of the Transformer is the self-attention mechanism, which enables the model to compute attention weights between all pairs of input positions. This allows the model to selectively focus on relevant spectral regions across the entire input. The self-attention operation is defined as:(14)$$\mathrm{Attention}\left(Q,K,V\right)=\mathrm{Softmax}\left(\frac{QK^{T}}{\sqrt{d_{k}}}\right)V$$
where *Q*, *K*, and *V* denote the query, key, and value matrices, respectively, and $d_{k}$ is the dimensionality of the key vectors used to scale the dot-product for numerical stability.

The resulting attention outputs are passed through stacked feed-forward networks (FFN) to model nonlinear relationships. In our implementation, the spectral input features are first projected into a unified embedding space, followed by multiple Transformer encoder layers to capture global feature interactions. A final regression head maps the encoded representations to the target quality indices such as moisture, oil, protein, and starch content.

Compared to traditional architectures such as CNNs or RNNs, the Transformer offers greater modeling flexibility and interpretability. Its ability to dynamically assign attention weights to important spectral regions enables the model to achieve superior prediction performance while providing insights into which wavelengths contribute most to the final output.

#### 2.2.5. CNN-Based Baseline Model

In this setting, each corn spectrum is represented as a one-dimensional signal of length 700, rather than a two-dimensional image. Thus, the convolution process applies directly along the spectral dimension.

As a comparative baseline, we also construct a Convolutional Neural Network (CNN) model (see Figure 3) for the regression prediction of multiple quality indicators from corn near-infrared (NIR) spectral data. CNNs are known for their strong local feature extraction capabilities, particularly suitable for identifying relevant patterns in structured input such as 1D spectral signals.

In this setting, the NIR spectrum is treated as a one-dimensional input sequence. The model architecture consists of multiple 1D convolutional layers, each followed by batch normalization and ReLU activation to ensure training stability and nonlinear representation. These layers learn localized filters to extract spectral features that may correspond to specific chemical compositions.

For example, if the kernel size is k=3, the convolution output at position *i* is computed as $y_{i}=w_{0}x_{i}+w_{1}x_{i+1}+w_{2}x_{i+2}+b$. This process is repeated across the entire spectral sequence, generating a feature vector (feature map). When multiple convolutional filters are applied in parallel, the resulting feature maps are stacked to form a feature matrix.

Formally, the output of a 1D convolutional layer can be expressed as:(15)$$y_{i}=\sigma \left(\sum_{j=0}^{k-1}w_{j}\cdot x_{i+j}+b\right)$$
where *x* is the input sequence, *w* denotes the convolution kernel of size *k*, *b* is the bias term, and σ(·) represents a nonlinear activation function such as ReLU. In practice, kernel sizes (e.g., 3, 5, 7) were selected to capture local absorption patterns of different widths, reflecting the fact that NIR spectra often contain both narrow peaks and broader bands. This operation captures local dependencies by applying the same kernel across the input sequence.

After convolutional processing, max-pooling layers are used to reduce dimensionality while preserving the most salient features. The resulting feature maps are flattened and passed through fully connected layers to produce the final regression outputs, including moisture, oil, protein, and starch content.

Compared with the Transformer model, the CNN emphasizes local dependencies and is efficient in capturing contiguous spectral patterns. However, it may be limited in modeling long-range interactions across distant spectral regions due to its constrained receptive field.

#### 2.2.6. MLP-Based Baseline Model

As another baseline, we introduce a Multi-Layer Perceptron (MLP) model to capture the global nonlinear relationships between input spectral features and the target quality indicators. The MLP consists of several fully connected layers stacked sequentially, each followed by nonlinear activation functions (e.g., ReLU) to introduce nonlinearity into the model.

The computation in each hidden layer is defined as:(16)$$h^{\left(l\right)}=\sigma \left(W^{\left(l\right)}h^{\left(l-1\right)}+b^{\left(l\right)}\right)$$
where $h^{\left(l-1\right)}$ is the input to the *l*-th layer, $W^{\left(l\right)}$ and $b^{\left(l\right)}$ denote the weight matrix and bias vector respectively, and σ(·) is an activation function such as ReLU.

In our task, the input layer receives the full-dimensional NIR spectral vector, and the output layer consists of multiple regression nodes corresponding to each quality indicator, including moisture, oil, protein, and starch content. The model is optimized by minimizing the Mean Squared Error (MSE) between predicted and true values.

While the MLP does not leverage local structure or sequential dependencies in the input spectrum, its fully connected architecture enables modeling of arbitrary global interactions across all spectral dimensions.

### 2.3. Data Preprocessing and Model Training

The training of the spectral Transformer network (SpecTran) is formulated as a multi-task regression problem, where the model simultaneously predicts multiple constituent concentrations (e.g., moisture, starch, oil, and protein) from a single near-infrared (NIR) spectrum. Let $D={\left\{\left(x_{i},y_{i}\right)\right\}}_{i=1}^{N}$ denote the training dataset, where $x_{i}\in R^{700}$ represents the preprocessed NIR spectrum with 700 wavelength channels, and $y_{i}\in R^{4}$ is the corresponding vector of ground-truth concentrations.

Given a model parameterized by θ, the predicted output is denoted as ${\hat{y}}_{i}=f_{\theta }\left(x_{i}\right)$. The training objective is to minimize a composite loss function that balances multiple objectives: prediction accuracy, task-specific importance, and regularization. We define the multi-task loss function as:(17)$$L\left(\theta \right)=\alpha \cdot L_{\mathrm{MSE}}+\beta \cdot L_{L1}+\gamma \cdot L_{\mathrm{weighted}},$$
where α, β, and γ are hyperparameters controlling the contribution of each component, and the individual loss terms are defined as follows.

The mean squared error (MSE) loss ensures global prediction accuracy:(18)$$L_{\mathrm{MSE}}=\frac{1}{N}\sum_{i=1}^{N}{∥{\hat{y}}_{i}-y_{i}∥}_{2}^{2}.$$

The L1 loss promotes sparsity and robustness to outliers:(19)$$L_{L1}=\frac{1}{N}\sum_{i=1}^{N}{∥{\hat{y}}_{i}-y_{i}∥}_{1}.$$

To address task imbalance—where some constituents (e.g., protein) are more challenging to predict than others—we introduce a weighted task-specific loss:(20)$$L_{\mathrm{weighted}}=\sum_{j=1}^{4}w_{j}\cdot \frac{1}{N}\sum_{i=1}^{N}{\left({\hat{y}}_{i,j}-y_{i,j}\right)}^{2},$$
where $w_{j}$ denotes the task-specific weight for the *j*-th constituent. In our experiments, we set w=[1.2,1.0,1.5,2.0] for moisture, starch, oil, and protein, respectively, based on empirical performance and relative prediction difficulty.

The overall loss in Equation (Equation 17) is minimized using the AdamW optimizer [13], which decouples weight decay from gradient computation, improving generalization:(21)$$\theta_{t+1}=\theta_{t}-\eta_{t}\left({\hat{m}}_{t}/\left(\sqrt{{\hat{v}}_{t}}+\epsilon \right)+\lambda \theta_{t}\right),$$
where $\eta_{t}$ is the learning rate at step *t*, ${\hat{m}}_{t}$ and ${\hat{v}}_{t}$ are bias-corrected estimates of the first and second moments of gradients, ϵ is a small constant, and λ is the weight decay coefficient.

To accelerate convergence and stabilize training, we employ the One-Cycle Learning Rate policy [14], which cyclically varies the learning rate between a minimum and maximum value over the course of training. Specifically, the learning rate increases linearly for the first 20% of epochs and then decreases cosine-annealed for the remaining 80%:(22)$$\eta_{t}=\eta_{\min}+\frac{1}{2}\left(\eta_{\max}-\eta_{\min}\right)\left(1+\cos\left(\pi \cdot \frac{t^{\prime }}{T}\right)\right),$$
where $t^{\prime }$ is the current step within the annealing phase, and *T* is the total number of annealing steps.

Additionally, gradient clipping is applied with a maximum norm of 1.0 to prevent gradient explosion during backpropagation through the deep transformer layers.

Training is performed for up to 250 epochs with early stopping: if the validation loss does not improve for 30 consecutive epochs, training is terminated. The model with the lowest validation loss is preserved for evaluation.

In summary, the proposed training strategy combines multi-task loss balancing, adaptive optimization, dynamic learning rate scheduling, and regularization techniques to achieve robust and accurate spectral regression, while the attention mechanisms provide interpretability through visualization of spectral importance.

## 3. Results

### 3.1. Model Comparison for Corn Quality Prediction

To evaluate the effectiveness of different models in corn quality prediction based on near-infrared (NIR) spectral data, we conducted a comprehensive comparison involving both traditional machine learning algorithms and advanced deep learning architectures. Specifically, we assessed nine representative models across four key quality traits—moisture, starch, oil, and protein—using a suite of evaluation metrics, including the coefficient of determination ($R^{2}$), root mean squared error (RMSE), and mean absolute error (MAE). The analysis aims to highlight each model’s strengths in capturing the underlying patterns of spectral data and their generalization capabilities across different traits. The results are visualized through radar and bar charts, followed by detailed tabular comparisons of prediction performance.

To comprehensively assess model performance in corn quality prediction, a radar chart was plotted to visualize the $R^{2}$ scores of nine models across four key traits: moisture, starch, oil, and protein, as shown in Figure 4. The chart compares both traditional machine learning models (e.g., PLS, Ridge, SVR, XGBoost) and neural network architectures (e.g., MLP, CNN, Transformer, and LSTTN). It is evident that the SpecTran model outperforms all others, achieving the highest $R^{2}$ values consistently, thus demonstrating its superiority in multi-trait prediction tasks.

To further compare model performance in terms of prediction error, we plotted the RMSE values across four corn quality indicators: moisture, starch, oil, and protein, as shown in Figure 5. The chart clearly illustrates that the SpecTran model achieves the lowest RMSE across all traits, with especially notable improvements in protein prediction, demonstrating its superior generalization capability.

Additionally, Table 2 and Table 3 summarize the average $R^{2}$ and RMSE scores of neural network and machine learning models, respectively. Table 4 further highlights detailed performance on moisture prediction, indicating that the SpecTran model yields the best accuracy among all candidates.

Table 2 presents the performance comparison of five neural network models on the corn quality prediction task. The evaluation metrics include average $R^{2}$, average RMSE, as well as the $R^{2}$ and RMSE values specifically for protein prediction. The SpecTran model achieves the best performance across all metrics, with an average $R^{2}$ of 0.483 and a protein $R^{2}$ of 0.368, outperforming the Transformer by 4.1 and 7.4 percentage points, respectively. In addition, the RMSE values are consistently lower, highlighting the model’s superior ability in learning complex feature representations.

Table 3 compares the performance of four commonly used machine learning models—PLS, Ridge, SVR, and XGBoost—with the proposed SpecTran model in corn quality prediction. The results indicate that traditional machine learning models generally underperform compared to deep neural networks. Among them, SVR yields the poorest performance with a protein $R^{2}$ of only 0.022 and a high RMSE of 0.641. XGBoost, the best among the machine learning methods, achieves an average $R^{2}$ of 0.409 and a protein $R^{2}$ of 0.291, both notably lower than those of SpecTran (0.483 and 0.368, respectively). These findings underscore the superior capability of deep models in capturing the complex relationships between spectral features and quality traits.

Table 4 presents a comparative analysis of moisture prediction performance across nine models, including both machine learning and neural network approaches. The results demonstrate that neural networks consistently outperform traditional methods in moisture estimation. Notably, the proposed SpecTran achieves the best performance on all three metrics, with an $R^{2}$ score of 0.635, RMSE of 0.173, and MAE of 0.140. In contrast, SVR exhibits the weakest performance with an $R^{2}$ of only 0.462 and relatively high errors. The trend highlights the strength of deep temporal models with attention mechanisms in capturing sequential dependencies within the spectral data for more accurate moisture prediction.

In summary, the comparative analysis clearly demonstrates the superior performance of the proposed SpecTran model in corn quality prediction. Across all evaluation metrics and quality traits, this model consistently outperforms both traditional machine learning algorithms and other deep neural networks. Its ability to capture complex temporal and spectral dependencies, coupled with the integration of attention mechanisms, contributes to its enhanced predictive accuracy. The results underscore the importance of designing domain-specific, temporally aware architectures when modeling agricultural quality traits using high-dimensional spectral data.

### 3.2. Spectral-Level Interpretability via Perturbation-Based Feature Importance Analysis

Understanding which spectral regions are most influential for predicting specific quality components is critical for enhancing both model interpretability and practical applicability. To this end, we conducted a detailed spectral-level feature importance analysis from two complementary perspectives: the first leveraging internal attention mechanisms embedded in the model, and the second employing a perturbation-based approach. These analyses aim to uncover the model’s internal decision-making patterns and elucidate how different wavelength regions contribute to the prediction of moisture, starch, oil, and protein content.

As shown in Figure 6 presents a detailed comparison of feature importance across the four primary quality components—moisture, starch, oil, and protein—based on the learned attention scores over spectral wavelengths for Sample 2 at epoch 100. Each colored curve represents the relative contribution of individual wavelengths to the prediction of a specific component.

Two prominent regions of high importance can be identified: the early spectral band between indices 0 and 100, and the later region around indices 580 to 660. In both regions, the protein curve (red) exhibits notably higher importance scores compared to the other components. This implies that the prediction of protein content is particularly sensitive to these spectral regions, possibly due to stronger NIR absorption characteristics of protein-related molecular bonds.

In contrast, the importance scores for moisture, starch, and oil appear more evenly distributed, with moderate peaks across the middle and high-wavelength bands. The overlapping but distinct importance profiles suggest that the model is capable of capturing component-specific spectral signatures, enabling multi-target learning with differentiated attention.

These results demonstrate the model’s ability to autonomously identify and emphasize critical wavelength regions relevant to each quality trait, which aligns with known spectral absorption features of organic compounds and validates the effectiveness of the attention-based learning mechanism.

To further reveal the decision-making mechanism of the model in predicting quality components, we conducted a perturbation-based feature importance analysis along the spectral wavelength dimension. Figure 7 presents the distribution of importance weights learned by the model across different wavelengths for four target components: moisture, oil, protein, and starch.

As shown in the figure, the feature importance distributions differ significantly among the components. For the protein component, two prominent peaks are observed in the spectral index ranges of approximately 0–100 and 580–660, indicating that the model focuses more on these specific wavelength bands when predicting protein content. In contrast, the importance curves for oil and moisture appear relatively smooth, without pronounced local peaks, suggesting that the model relies on a more distributed spectral basis for their prediction. The starch component, on the other hand, shows a moderate attention region within the mid-wavelength range (approximately 300–450).

These differentiated attention patterns demonstrate the model’s strong capability to distinguish informative spectral bands and to automatically capture discriminative regions, thereby improving both prediction accuracy and interpretability. Notably, the high-attention spectral regions align with several known absorption bands in near-infrared spectroscopy, such as the N–H combination bands around 6900–7100 cm^−1^ and the C–H overtone bands near 5600–5800 cm^−1^ [15,16], implying that the model implicitly learns underlying domain knowledge during training. This wavelength-level interpretability not only enhances the trustworthiness of the model but also provides theoretical guidance and practical references for sensor design and representative wavelength selection.

To gain a deeper understanding of how the model utilizes spectral features for quality component prediction, we further examined a single-sample interpretability case, as shown in Figure 8. This figure presents the perturbation-based feature importance curves for Sample 20, comparing the learned wavelength-wise importance scores (in red) with the original absorbance spectra (in blue) across four key quality traits: moisture, starch, oil, and protein.

Distinct component-specific attention patterns can be observed. For instance, both the protein and oil components exhibit sharp importance peaks in the spectral index range of approximately 500 to 700, indicating that the model focuses strongly on these regions when predicting these two traits. This behavior aligns with known absorption characteristics in near-infrared spectroscopy (NIRS), where protein and lipid-related bonds often show stronger signals in the higher-wavelength bands.

In contrast, the moisture and starch components demonstrate a relatively smoother and more distributed importance profile, suggesting that the model relies on a broader range of wavelengths for their prediction. Such differences in spectral dependency highlight the model’s ability to distinguish wavelength regions that are informative for different targets, thereby enabling a form of implicit spectral feature selection.

Overall, the results illustrated in Figure 8 support the model’s capability to extract interpretable, component-specific insights from the spectral data, reinforcing both its prediction performance and its transparency for downstream applications such as sensor optimization and band selection.

In summary, the spectral-level feature importance analysis reveals that the model exhibits component-specific attention patterns that are not only internally consistent across methods but also consistent with known spectral absorption characteristics in NIR spectroscopy. This consistency enhances the model’s trustworthiness and scientific relevance. Furthermore, such insights may guide the optimization of spectral sensor design and wavelength selection, paving the way for more efficient, interpretable, and application-oriented spectral prediction systems.

## 4. Discussion

To gain further insight into the internal reasoning process of the model, we examine the attention weight distribution from the final Transformer layer (Layer 6) for Sample 20, as visualized in Figure 9. This heatmap captures the pairwise attention intensities between query and key positions across the spectral sequence.

A prominent observation from the heatmap is the strong attention concentration in the final positions of the sequence (around index 41 and 42), both as query and key elements. These positions receive disproportionately high attention weights, as indicated by the yellowish blocks in the bottom-right region of the heatmap. This suggests that the model tends to aggregate high-level contextual information toward the end of the sequence, possibly treating the terminal tokens as semantic summary anchors.

Our model demonstrates superior performance compared to many evaluated baselines, including recent methods [7,17]. However, we acknowledge a key limitation in moisture detection, where our performance did not surpass specialized algorithms like that of Lin et al. [18]. We hypothesize this is a trade-off: SpecTran’s multi-scale architecture (see Section 2.2.2) is optimized for complex, non-linear spectral patterns, which may be less effective for the dominant, localized signals characteristic of moisture. This suggests a performance ceiling for moisture detection within the NIR modality itself. Indeed, as noted by Tao et al. [19], alternative non-destructive methods, such as microwave-based sensing, have shown superior performance (e.g., in $R^{2}$ and RMSE) for this specific application.

Moreover, moderate attention activations can also be found near intermediate positions (e.g., around position 23), reflecting the model’s capability to distribute focus over both global and local features. Such a hierarchical attention pattern aligns with the intuition that earlier layers capture low-level spectral transitions, while deeper layers consolidate broader context relevant to the final prediction.

This attention pattern confirms the model’s ability to learn task-specific dependencies. It also enhances interpretability by clearly identifying the spectral regions that dominate final predictions. These insights could further guide model pruning, feature selection, or the design of lightweight attention mechanisms for deployment scenarios.

Beyond attention-based interpretability, the model also demonstrates strong predictive performance. As presented in earlier sections, the model consistently outperformed baseline approaches across all four quality traits—moisture, protein, starch, and oil. Notably, the improvements were particularly significant for protein and oil, indicating the model’s superior capacity to capture complex spectral dependencies relevant to these components.

This observation aligns closely with the perturbation-based feature importance results, where spectral regions in the 500–700 wavelength range were shown to have elevated importance scores for protein and oil prediction. Such correspondence validates that the model is effectively attending to chemically meaningful spectral bands rather than relying on spurious correlations.

Furthermore, attention maps across different samples and layers reveal that the model dynamically shifts its focus. Rather than uniformly distributing attention across the input, the model learns to allocate attention based on the task-specific and sample-specific spectral patterns. This reinforces the model’s interpretability and adaptability in real-world multi-component prediction tasks.

Beyond empirical performance and interpretability insights, the analysis also sheds light on the two fundamental challenges faced in NIR-based corn quality prediction. First, the attention-based visualizations confirm the model’s nonlinear reasoning capability—evidenced by its ability to dynamically emphasize chemically meaningful spectral regions—which addresses the limitation of prior linear or shallow models in capturing complex absorbance-to-composition mappings. Second, the observed multi-head, multi-layer attention patterns reflect the model’s competence in learning hierarchical, multi-scale dependencies across the spectral sequence. This property is crucial for extracting informative features that span both local peaks and global wavelength interactions, which are often entangled in corn’s heterogeneous biochemical matrix. These findings substantiate SpecTran’s design rationale and offer a promising direction for future work in lightweight attention calibration, spectrum-specific prior integration, and model pruning for field deployment scenarios.

Overall, these findings substantiate the design choices of incorporating temporal modeling and attention mechanisms, and provide a theoretical grounding for the model’s robust performance and generalization ability in spectral quality prediction tasks.

## 5. Conclusions

This study addresses two critical scientific challenges in corn quality prediction using near-infrared (NIR) spectroscopy: (1) the limited ability of traditional models to capture the complex nonlinear relationships between spectral absorbance and quality traits such as protein and oil content; and (2) the lack of multi-scale contextual modeling across spectral bands, which hinders the extraction of deep and informative features. To overcome these limitations, we propose SpecTran—a Transformer-based spectral modeling framework that integrates multi-scale patch embedding, spectrally enhanced positional encoding, and hierarchical attention mechanisms. This architecture enhances the model’s ability to simultaneously learn both global dependencies and local discriminative features within the spectral sequence.

Experimental results on the publicly available Eigenvector corn dataset demonstrate that SpecTran achieves superior predictive performance across four key quality indicators—moisture, protein, starch, and oil—with an average $R^{2}$ of 0.483. Specifically, the root mean squared error (RMSE) for protein and oil prediction was reduced by 11.2% and 10.7%, respectively, compared to strong baselines, validating the model’s effectiveness in modeling nonlinear spectral relationships. Moreover, interpretability analysis via attention heatmaps and perturbation-based attribution confirms that the model consistently attends to chemically meaningful wavelength regions, enhancing both reliability and scientific insight.

In summary, SpecTran provides a robust and interpretable solution for NIR-based corn quality assessment, offering significant improvements in accuracy and generalizability. This work lays a solid foundation for future research in agricultural spectroscopy and supports practical deployment in food safety and grain grading applications.

These findings carry substantial practical and scientific implications. From an agricultural and industrial perspective, SpecTran’s enhanced accuracy and interpretability enable more reliable, rapid, and cost-effective quality screening of corn batches, supporting better decisions in grain trading, feed formulation, and food processing. The model’s ability to highlight chemically relevant spectral bands (e.g., 500–700 nm for protein and oil) also provides actionable guidance for sensor design and band selection, paving the way for compact, low-cost NIR devices tailored to specific quality traits.

From a methodological standpoint, this work demonstrates that domain-aware architectural innovations—such as multi-scale spectral embedding and spectrally enhanced positional encoding—can significantly improve the performance of Transformer models on small-scale, high-dimensional spectroscopic data. This challenges the prevailing notion that deep learning requires massive datasets and offers a template for developing interpretable, physics-informed AI models in other spectroscopic domains, including food safety, pharmaceuticals, and environmental monitoring.

## Figures and Tables

**Figure 1 foods-14-03786-f001:**
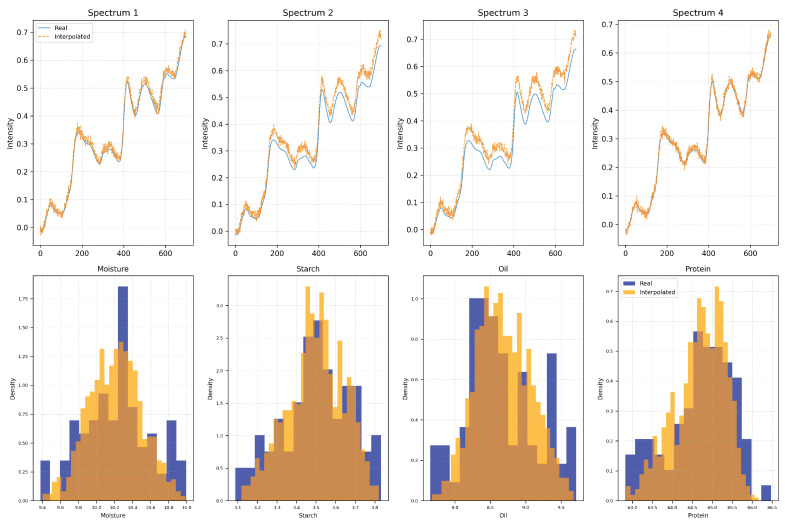
Visualization of Spectrum and Component Distributions Before and After Augmentation. The top row shows the comparison between original (blue) and interpolated (orange) spectra for four randomly selected samples, demonstrating the smoothness and continuity of the generated spectral curves. The bottom row presents the density distributions of four quality components (moisture, starch, oil, and protein) before (blue) and after (orange) augmentation. The augmented samples exhibit similar distributional characteristics with slight variance enhancement, indicating that the proposed interpolation-based strategy can effectively enrich the training set while maintaining distribution fidelity.

**Figure 2 foods-14-03786-f002:**
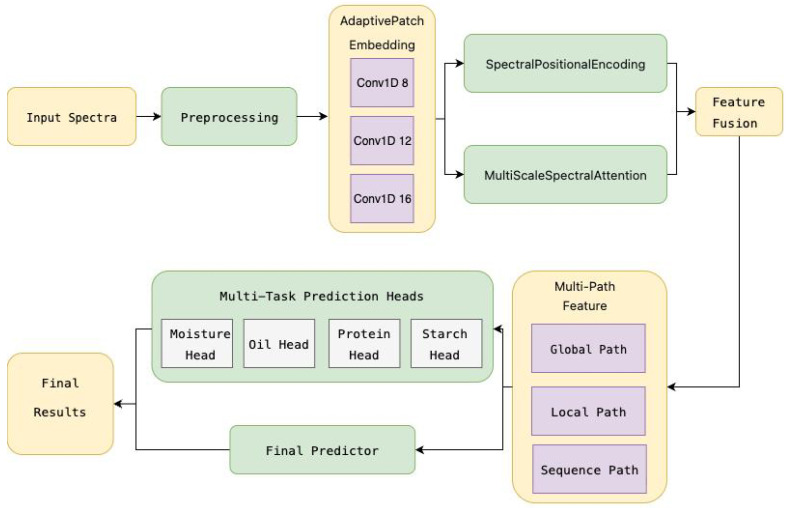
Architecture of the SpecTran model for multi-task NIR spectral regression. The input spectrum undergoes preprocessing and is then processed through an adaptive patch embedding layer with multiple kernel sizes (8, 12, 16). Spectral positional encoding and multi-scale spectral attention enhance feature representation. The resulting sequence is fused via three parallel pathways: global, local, and sequential, before being fed into task-specific prediction heads and a shared final predictor. The final output combines specialized and generalized predictions to improve accuracy and robustness.

**Figure 3 foods-14-03786-f003:**
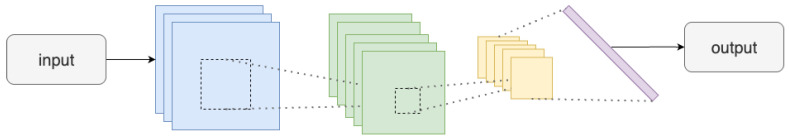
Architecture of the CNN-based baseline model for multi-task NIR spectral regression.

**Figure 4 foods-14-03786-f004:**
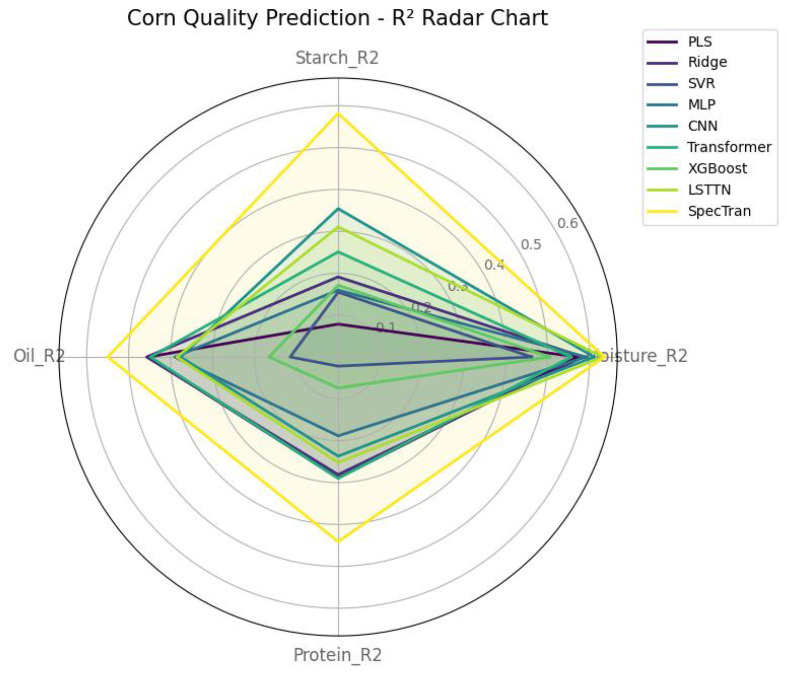
Radar chart of $R^{2}$ scores for different models in corn quality prediction. The figure compares nine models (including machine learning and neural networks) across four quality traits: moisture, starch, oil, and protein. The SpecTran model consistently outperforms other methods, achieving the best $R^{2}$ values on all metrics.

**Figure 5 foods-14-03786-f005:**
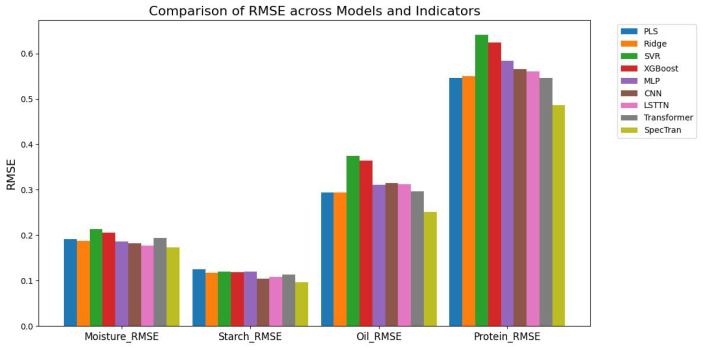
Bar chart of RMSE values across different models for corn quality prediction. The chart compares nine models on four indicators: moisture, starch, oil, and protein. Overall, the SpecTran model achieves the lowest RMSE across all traits, indicating superior prediction accuracy.

**Figure 6 foods-14-03786-f006:**
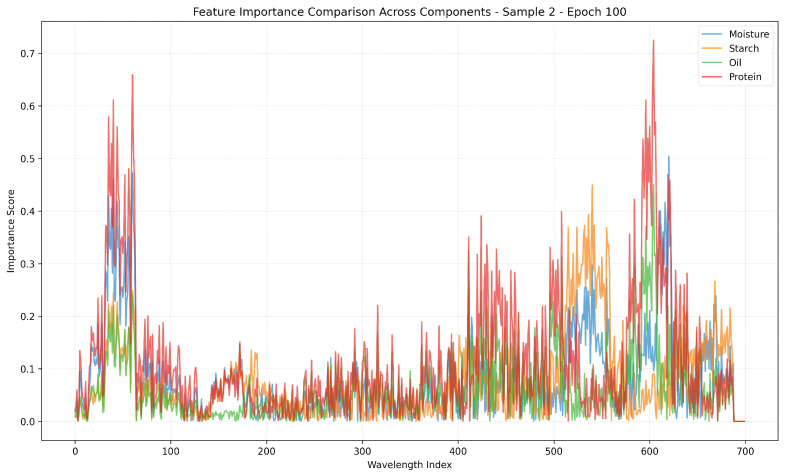
Feature importance comparison across quality components along spectral dimensions (Sample 2, Epoch 100). This figure illustrates the wavelength-wise feature importance learned by the model at training epoch 100 for Sample 2. It compares the importance scores of four target components: moisture, starch, oil, and protein. Notably, distinct peaks are observed in the 0–100 and 580–660 spectral regions, where the protein component exhibits significantly higher importance values. This indicates that the model pays more attention to specific wavelengths when predicting protein content, shedding light on the model’s attention mechanism and spectral selection behavior.

**Figure 7 foods-14-03786-f007:**
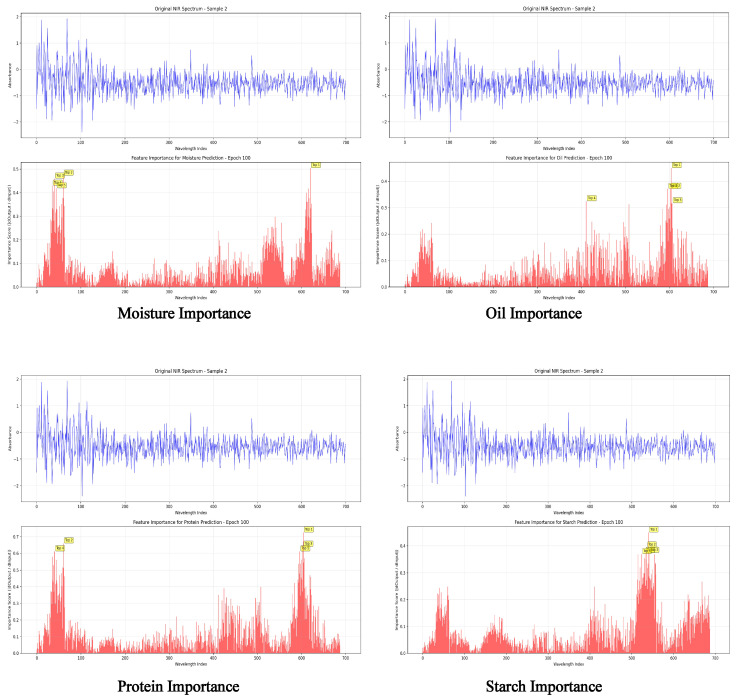
Feature importance distribution across spectral dimensions for four quality components (Sample 2, Epoch 100). This figure visualizes the spectral-wise feature importance scores obtained through perturbation-based analysis at epoch 100 for Sample 2. The plots compare the contribution of different wavelength indices to the model’s predictions of moisture, oil, protein, and starch content. Notably, component-specific attention patterns emerge—for instance, the protein component shows dominant importance in the 0–100 and 580–660 spectral regions. These findings highlight the spectral sensitivity of the model and support its interpretability in component-specific prediction tasks.

**Figure 8 foods-14-03786-f008:**
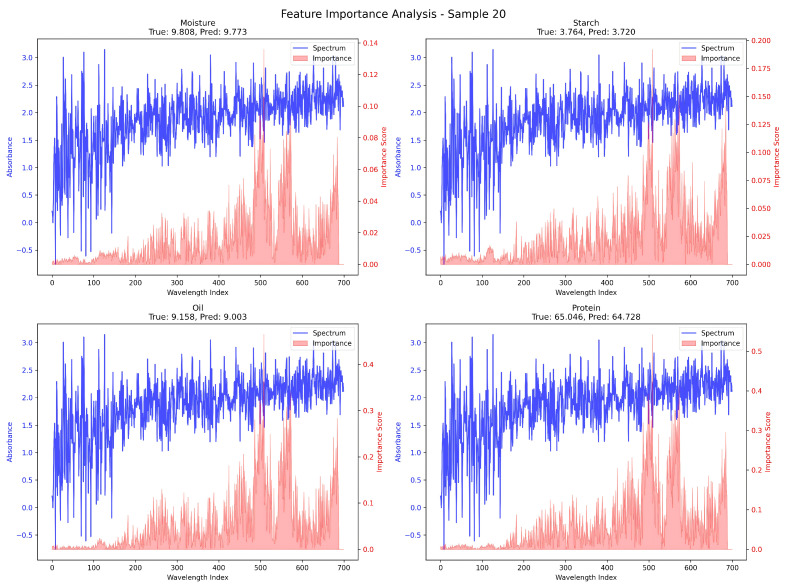
Spectral Feature Importance Analysis for Sample 20. This figure illustrates the perturbation-based feature importance analysis along the spectral dimension for Sample 20 across four quality components: moisture, starch, oil, and protein. The blue curve represents the original absorbance spectrum, while the red curve indicates the importance score of each wavelength index. Notably, the importance scores for protein and oil rise significantly in the 500–700 wavelength range, suggesting that the model relies more heavily on these regions for accurate prediction. In contrast, moisture and starch exhibit more dispersed and less pronounced attention patterns. This analysis highlights the model’s selective attention mechanism over spectral bands and enhances interpretability in component-wise prediction.

**Figure 9 foods-14-03786-f009:**
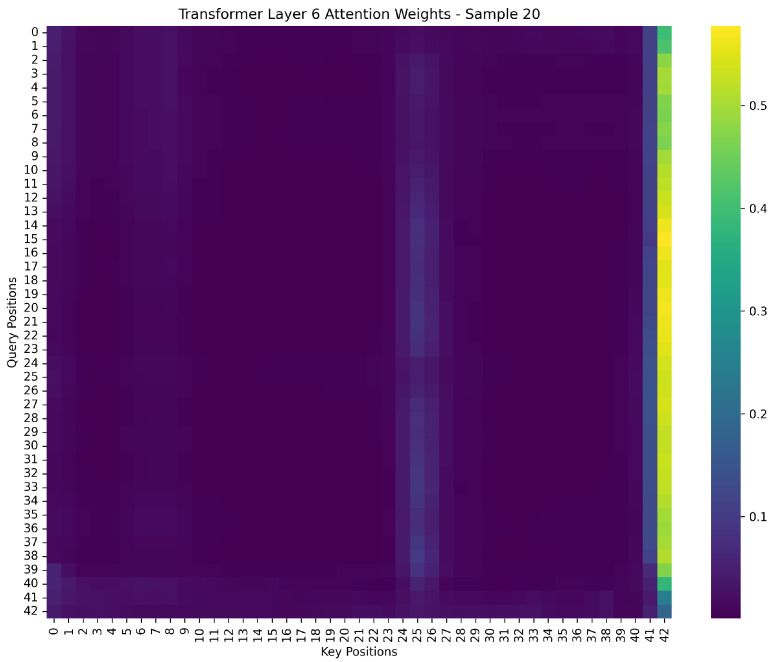
Transformer Layer 6 Attention Weights for Sample 20. This heatmap visualizes the attention weight distribution of the sixth transformer layer when processing Sample 20. The x-axis denotes the key positions, and the y-axis represents the query positions. Brighter colors indicate higher attention weights. A strong concentration of attention is observed near the final sequence positions (around position 41 and 42), suggesting that the model heavily relies on these terminal tokens for high-level semantic summarization and final prediction. This pattern demonstrates the model’s ability to focus on contextually important information in deeper layers.

**Table 1 foods-14-03786-t001:** Descriptive statistics of the four biochemical constituents in the Eigenvector corn dataset.

Constituent	Sample Count	Min	Max	Mean	Std
Moisture (%)	80	9.377	10.993	10.234	0.380
Starch (%)	80	3.088	3.832	3.498	0.177
Oil (%)	80	7.654	9.711	8.668	0.499
Protein (%)	80	62.826	66.472	64.696	0.821

**Table 2 foods-14-03786-t002:** Performance comparison of neural network models.

Model	Avg_R2	Avg_RMSE	Protein_R2	Protein_RMSE
MLP	0.332	0.300	0.188	0.584
CNN	0.395	0.292	0.237	0.566
LSTTN	0.395	0.290	0.252	0.560
Transformer	0.442	0.278	0.294	0.537
**SpecTran**	**0.483**	**0.265**	**0.368**	**0.517**

**Table 3 foods-14-03786-t003:** Performance comparison of machine learning models.

Model	Avg_R2	Avg_RMSE	Protein_R2	Protein_RMSE
PLS	0.348	0.289	0.289	0.547
Ridge	0.378	0.287	0.281	0.550
SVR	0.188	0.337	0.022	0.641
XGBoost	0.409	0.279	0.291	0.540
**SpecTran**	**0.483**	**0.265**	**0.368**	**0.517**

**Table 4 foods-14-03786-t004:** Moisture prediction performance comparison of machine learning and neural network models.

Model	Moisture_R2	Moisture_RMSE	Moisture_MAE
PLS	0.571	0.191	0.155
Ridge	0.585	0.188	0.154
SVR	0.462	0.214	0.172
XGBoost	0.598	0.185	0.151
MLP	0.592	0.186	0.150
CNN	0.611	0.182	0.145
LSTTN	0.618	0.179	0.143
Transformer	0.624	0.177	0.142
**SpecTran**	**0.635**	**0.173**	**0.140**

## Data Availability

The corn NIR spectroscopy dataset used in this study is available in the GitHub repository at https://github.com/Loewen-Hob/NIR-Corn.git (accessed on 31 October 2025). This repository contains the raw spectral data, reference constituent values (moisture, starch, oil, and protein), preprocessing scripts, and model implementation code. The data are provided in open formats (CSV, NumPy) for easy access and reproducibility. The dataset is derived from the public Eigenvector Research corn dataset [4], with additional processing and split strategies applied as described in the manuscript.

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
