# Peer review of "Multi-Path Attention Fusion Transformer for Spectral Learning in Corn Quality Assessment"

_foods, 2025, doi:10.3390/foods14213786_

Round 1
Reviewer 1 Report
Comments and Suggestions for Authors
Please use a more intuitive, simpler title.
Need more information about the dataset used.
What is the size of an image? (length and width?)
Need more detail about the experimental setup as to how the spectral and content (protein, oil, etc.) data were obtained.
Does an image fully contain a corn sample, without background? If so, was the camera placed at a constant distance for all corn samples? Were the corn samples approximately of equal size?
How many pixels are there in an image?
Each pixel will have the spectral data, as such does each pixel have content (i.e., protein, oil, starch) data as well?
The above information needs to be clarified before explaining the convolution process.
As it is presented now, the convolution process is not clear.
Need justification as to how the kernel (k) sizes were selected--to justify this, clear information about the images including their dimensions and number of pixels is required.
Better to do a graphical illustration of how the convolution works--in equation one, it is not clear to me as to how the output of a 1D convolution ends up being a matrix.
Need more granular and clearer explanations at such points--maybe first explain how this is done to one pixel, then say the process is repeated for all pixels and that can result in a matrix.
Reviewer 2 Report
Comments and Suggestions for Authors
General:
The manuscript presents an interesting and timely study proposing a spectral Transformer network (SpecTran) for modeling near-infrared (NIR) spectra to predict corn silage quality. This approach represents an advancement over traditional chemometric methods such as Partial Least Squares Regression (PLSR), Principal Component Regression (PCR), and Convolutional Neural Networks (CNNs), which have limitations—especially when modeling heterogeneous samples like corn silage. The study compares SpecTran against several commonly used models across four key corn silage quality parameters: moisture, starch, oil, and protein.
The authors clearly describe the architecture of SpecTran, its novel features tailored for NIR spectral regression, and demonstrate its advantages over existing methods.
Major Comments and Suggestions
- Manuscript Length and References:
- The manuscript is relatively long (19 pages), with detailed descriptions of the materials and methods and results sections. While thoroughness is appreciated, some sections could be streamlined for conciseness and better flow.
- Only 15 references are cited, which seems limited given the scope and contemporary nature of the work. Including more recent and relevant literature would strengthen the manuscript’s foundation and contextualize it within current research trends.
- Tables and Figures:
- Each table should be fully self-explanatory. Please include a list of all abbreviations and symbols directly below each table for ease of understanding by readers.
- Abstract:
- The abstract effectively identifies the core problem in NIR-based corn quality prediction and presents SpecTran as a novel solution. It highlights key technical components and strong experimental results, including quantitative metrics, and emphasizes interpretability alongside accuracy.
- Suggestions:
- The first sentence is quite long and dense. Consider splitting it into two sentences to enhance readability.
- Specify what the “strongest baseline” refers to when discussing performance comparisons.
- Simplify or briefly explain technical terms such as “adaptive multi-scale patch embedding” and “spectral-enhanced positional encoding” for a broader audience.
- Introduction:
- The introduction provides a clear narrative tracing the evolution from traditional linear models to modern deep learning techniques in NIR spectral analysis, culminating in a strong rationale for the proposed model.
- However, several sentences are overly long and contain nested clauses, which can hinder comprehension. Breaking these into shorter, clearer sentences will improve readability.
- Phrases like “nonlinear mapping” and “multi-scale contextual dependencies” appear repeatedly. Rephrasing these terms or using synonyms will help maintain reader engagement.
- Lines 77-80 contain a sentence that reads more like a summary or conclusion and should be removed from the introduction. Instead, this section would benefit from a clear statement of the manuscript’s hypothesis and specific research objectives.
- Materials and Methods:
- The dataset description lacks basic descriptive statistics (minimum, maximum, mean, standard deviation) for the four biochemical constituents (moisture, starch, oil, protein). Including these statistics is important for understanding the data distribution and the complexity of the regression problem.
- It is unclear whether the spectra of corn silage obtained for further processing originate from corn samples that were scanned fresh or dried/ground. Clarification is essential as sample preparation affects spectral data.
- The dataset consists of only 80 samples, which is quite small for training and validating complex deep learning models. This limitation raises concerns about potential overfitting, generalizability of the results, and statistical robustness. These limitations should be acknowledged and discussed.
- Aside from these points, the methods section is well-documented, supported by relevant equations and clear descriptions.
- Discussion:
- The discussion section repeats certain ideas multiple times, such as the model’s ability to focus on chemically meaningful spectral regions. Consider consolidating these repetitions for conciseness.
- Some sentences are lengthy and awkwardly constructed. Simplifying sentence structure will enhance the flow and readability of this section.
- Conclusion:
- The conclusion is clear, well-structured, and effectively summarizes the study’s contributions.
Review summary
This manuscript presents a valuable contribution by introducing SpecTran for enhanced NIR spectral modeling of corn silage quality. Addressing the above points—particularly dataset description, manuscript conciseness, and clarity—will significantly improve the manuscript’s quality and accessibility.
Reviewer 3 Report
Comments and Suggestions for Authors
Near–infrared spectra for predicting food composition is a recurring and well-established topic. Acceptable levels of MAE or RMSE for protein or fat assessed by NIR should be lower than 2% and higher than 0.85 in terms of R2. The abstract suggest that Spectran had an R2 of 0.987, this result is reiterated in the introduction, and it is misleading. However, tables and figures show values lower than 0.7. The manuscript requires complete description of methods. Other comments and suggestions are:
line 37 "their innate ability"
their ability
line 38 "1D spectral data"
one-dimensional spectral data
Line 104
Figure 1.
All figures and tables must be cited in the text and should appear at the end of the paragraph in which they are first mentioned.
Line 111. All symbols, abbreviations in equations must be explicitly defined. I recommend using a different font type when citing Pytorch functions for example, Softmax. Here, Zm are "patch sequences"? what is m? What is Conv1D, is it Conv1d?
Line 180 "a Convolutional Neural Network (CNN)"
The CNN model is the base line, then why in the results section Figure 3 shows "different models"? Redraft this section to fully justify and describe what is going to be presented in the results section.
Line 256. Describe computing set up, software and hardware. Model testing and validation requires data spliting, please describe your procedures to test model performance.
Line 262 "To validate the effectiveness of this approach"
Provide statistical evidence. The plots are not sufficient evidence.
Why this procedure was needed?
I recommend sub-section 3.1 to be moved to the Methods section.
Line 271. All description of methods, procedures must be presented in the Methods section, redraft subsection section 3.2, and present here only the results.
Line 286 " is evident that the SpecTran model outperforms all others, achieving the highest R2 values consistently"
Why were these models used?
Please review the literature on machine learning modeling of NIR data, justify your selection of models based on this review, and/or identify the best-performing algorithms to compare against SpecTran.
Figure 3. Trait labels should be Protein instead of Protein_R2. Line 289. Radar charts looks good and are easy to interpret. Present a figure with all three goodness of fit measures R2, RMSE and MAE. Line 294 Delete table 1,2 and 3.
Line 340 Present Figure 5 with individual plots to improve clarity in the identification of spectral regions. Probably figure 6 is sufficient. Please explain why is important to consider/evaluate sample 2 in figure 5 or sample 20 in figure 6.
Discussion needs improvement. Some questions that could be answered by contrasting the results to other findings reported in the literature are:
* How compares the performance of the SpecTran to other models? For instance Cui et al. (2024) or Fatemi et al. (2022)
https://doi.org/10.3390/foods13111722
https://doi.org/10.1016/j.foodchem.2022.132442
* Limitations of the study and the database. Particularly please elaborate on the definition of protein and fat. For example, protein is crude protein (N x 6.25)? Please provide descriptive statistics of corn composition. Are samples 2 and 20 considered typical? Why is it important to examine the spectra of these particular samples?
* The significance and implications of key findings.
Round 2
Reviewer 3 Report
Comments and Suggestions for Authors
Dear Authors,
Here I provide several comments and suggestions aimed at improving the quality of the manuscript. In its current form, there are several aspects that require further attention.
Line 12 — An R² value of 0.483 cannot be considered state-of-the-art when traditional methods have achieved an R² of 0.9604, as indicated in line 30. The authors’ conclusion (line 14) is not supported by the evidence presented in the study. There are numerous studies in the literature analyzing the spectral characteristics of grains to determine their properties. For example, Lin et al. (2019) proposed a procedure to estimate the moisture content of rice grains and achieved an R² of 0.93. The authors may consider examining why their own procedure does not perform as effectively. https://doi.org/10.3390/app9081654
Line 76 "The corn near-infrared (NIR) spectral dataset used in this study was obtained
from the publicly available repository hosted by Eigenvector Research (http://www.
eigenvector.com/data/Corn )."
The corn near-infrared (NIR) spectral dataset used in this study was obtained
from the publicly available repository hosted by Eigenvector Research [6].
Line 108 "which is detailed in Section 2.2.1."
Line 113 "interpolating between existing samples"
There are many interpolation methods, and the description provided is insufficient.
Does "data augmentation" mean increasing the number of samples? Please clearly explain how the continuity and smoothness of the interpolated spectrum (the ORANGE line in Figure 1) preserve the structural features of the original spectrum. It appears that the interpolated data correspond to the BLUE line.
Delete figure 4 and 5. The result is clearly presented in Table 2.
Line 299 "In summary, the proposed training strategy"
Line 545 "with additional processing and split strategies applied as described in the manuscript."
Line 331 "the performance comparison of five neural network models on the
corn quality prediction task"
Please describe the data-splitting strategies in detail: how many original samples were used for training, and how many interpolated samples were generated? Was the model trained exclusively on interpolated data? Was only a single split performed, or were multiple splits used (five)? Additionally, clarify how the five models for each algorithm were generated. Finally, note that "corn quality" is not equivalent to "corn composition".
Line 339 "in corn quality prediction"
in corn protein content prediction
Table 2 Clearly state that Avg_R2 and Avg_RMSE are the average not only of five models but four (or three?, does this average exclude protein?) corn characteristics.
350 "outperform traditional methods in moisture estimation"
Does this statement refer specifically to machine learning estimation procedures, or does it also include traditional methods such as the gravimetric determination of moisture content?
Round 3
Reviewer 3 Report
Comments and Suggestions for Authors
The authors, in their response to Comment 1 by Reviewer 3, acknowledge that the original phrasing in line 30 was misleading. While their response to the comment is adequate, the corresponding revision was not incorporated into the updated version of the manuscript. Therefore, my recommendations are:
Line 10 "Evaluated on the public Eigenvector corn dataset, SpecTran achieves state-of-the-art performance across four key traits—moisture, starch, oil, and protein—with an average R2 of 0.483. It reduces RMSE by 11.2% for protein and 10.7% for oil compared to the best-performing baseline, which is the standard Transformer model."
change to:
Evaluated on the public Eigenvector corn dataset, SpecTran had a performance across four key traits—moisture, starch, oil, and protein—with an average R2 of 0.483. It reduced RMSE by 11.2% for protein and 10.7% for oil compared to the best-performing baseline, which is the standard Transformer model."
In the Discussion section, please address why the authors’ procedure did not perform as effectively as other algorithms, particularly that of Lin et al. (2019) in relation to moisture detection. This request was made in a previous revision but was not incorporated into the current version of the manuscript. Additionally, the recent work by Tao et al. (2025) provides an important overview of non-destructive microwave-based testing methods that demonstrate better performance in terms of R2 and RMSE (see Table 2 in Tao et al., 2025).
https://doi.org/10.3390/app9081654
https://doi.org/10.3390/s25154783
Comments on the Quality of English Language
.
